**PLOS** COMPUTATIONAL BIOLOGY

# OSS-DBS: Open-source simulation platform for deep brain stimulation with a comprehensive automated modeling

Konstantin Butenko[1]*, Christian Bahls[1], Max Schröder[2], Rüdiger Köhling[3,4], Ursula van Rienen[1,5]

**1** Institute of General Electrical Engineering, University of Rostock, Rostock, Germany, **2** Institute of Communications Engineering, University of Rostock, Rostock, Germany, **3** Oscar-Langendorff-Institute of Physiology, Rostock University Medical Center, Rostock, Germany, **4** Interdisciplinary Faculty, University of Rostock, Rostock, Germany, **5** Department Life, Light & Matter, University of Rostock, Rostock, Germany

\* konstantin.butenko@uni-rostock.de

## Abstract

In this study, we propose a new open-source simulation platform that comprises computer-aided design and computer-aided engineering tools for highly automated evaluation of electric field distribution and neural activation during Deep Brain Stimulation (DBS). It will be shown how a Volume Conductor Model (VCM) is constructed and examined using Python-controlled algorithms for generation, discretization and adaptive mesh refinement of the computational domain, as well as for incorporation of heterogeneous and anisotropic properties of the tissue and allocation of neuron models. The utilization of the platform is facilitated by a collection of predefined input setups and quick visualization routines. The accuracy of a VCM, created and optimized by the platform, was estimated by comparison with a commercial software. The results demonstrate no significant deviation between the models in the electric potential distribution. A qualitative estimation of different physics for the VCM shows an agreement with previous computational studies. The proposed computational platform is suitable for an accurate estimation of electric fields during DBS in scientific modeling studies. In future, we intend to acquire SDA and EMA approval. Successful incorporation of open-source software, controlled by in-house developed algorithms, provides a highly automated solution. The platform allows for optimization and uncertainty quantification (UQ) studies, while employment of the open-source software facilitates accessibility and reproducibility of simulations.

## Author summary

Volume conductor models for the computation of the potential and current distribution resulting from deep brain stimulation can help research to gain a deeper understanding of the underlying processes as well as in optimization studies. On the other hand, they are extremely valuable for patient-specific therapy planning while avoiding side effects as far as possible. Despite existing high-quality models, further potential exists to increase their

**Data Availability Statement:** All files related to the submitted manuscript are available at https://github.com/SFB-ELAINE/OSS-DBS.

**Funding:** This work and the authors are funded by the Deutsche Forschungsgemeinschaft (DFG, German Research Foundation) – SFB 1270/1 - 299150580. The funders had no role in study design, data collection and analysis, decision to publish, or preparation of the manuscript.

**Competing interests:** The authors have declared that no competing interests exist.

level of realism, precision and reliability and to allow robust optimization. Our approach enables high-precision, patient- or atlas-based results for deep brain stimulation while simultaneously exploiting different measures to achieve high computational efficiency. In the development of the simulation software, we follow the goals of Open Science—in particular the principles of open-source, open data and reproducibility. In two benchmark examples, one on the human brain, the other on the rat brain, we were able to clearly demonstrate the accuracy and efficiency of our simulation results in comparison to a high-resolution simulation using a commercial software. The developed platform provides both the scientific community and clinicians with a precise yet easy-to-use simulation tool for scientific optimization studies and patient-specific therapy planning in context of deep brain stimulation.

This is a *PLOS Computational Biology* Software paper.

## Introduction

Deep brain stimulation (DBS) is one of the most important treatment options for patients with Parkinson's Disease (PD) or dystonia, superior to pharmacological treatment alone regarding PD, and often the only option for patients with dystonia [1]. While the therapeutic effect is thought to arise from a normalization of pathological discharge pattern [2] consisting of accentuated $\beta$-oscillatory activity in PD patients [3, 4], and persistent $\alpha$-band activity at rest in dystonia patients [5], the mechanism of this effect is still under debate. The basic principle of this therapy is to deliver electric pulses into basal ganglia nuclei via an electrode. A rectangularly shaped signal with a repetition rate of 130 Hz and a pulse duration of 60-90 $\mu$s is the most common clinically applied stimulation protocol, though numerous modifications were suggested. Importantly, the volume of tissue activated (VTA) appears to be one of the most substantial factors defining the therapeutic success of DBS [1]. This, and the fact that indeed neither motor nor non-motor effects of DBS are understood is a clear indication that the response of neural networks to DBS requires further investigation in order to increase the efficiency of the treatment.

An accurate numerical estimation of the electric field distribution and corresponding neural activation is an essential step towards DBS optimization that will define an optimal electrode geometry, its location, and a stimulation protocol. Moreover, employment of patient-specific *in silico* models can guide therapy planning by predicting therapeutic and side effects. The validity of such models strongly depends on how detailed the Volume Conductor Model (VCM) is. In particular, the electric field distribution is strongly influenced by the conductivity of the tissue, which has a heterogeneous, locally anisotropic, and dispersive nature. The latter factor necessitates field evaluations over a wide range of frequencies. Furthermore, the capacitive properties of grey and white matter cannot be disregarded for current-controlled stimulations [6].

The open-source simulation platform OSS-DBS, proposed in this work, takes into account the aforementioned aspects and enables a problem-specific mesh convergence analysis, based on solving the electro-quasistatic (EQS) problem. Moreover, implemented automated routines simplify the process of numerous calculations, normally required by optimization and UQ

algorithms. The platform accommodates fast reproducibility of the results by other research groups: the combination of publicly-accessible computer-aided design (CAD) and computer-aided engineering (CAE) modules, connected and controlled via a Python interface, facilitates rapid setup of VCMs. The platform was primarily developed for use among experimental researchers, and thus requires only a relatively simple presetting of input parameters via a graphical user interface. This work originates from [7], where an open-source workflow for electric field calculation was described. However, in that paper the authors' primary aim was to evaluate neural activation extent and field thresholds. Although, the platform adopts partially the proposed workflow, this study is more focused on creating realistic volume conductor models.

## Design and implementation

OSS-DBS is comprised of a number of modules, which are required for generation and discretization of the computational domain, its physics-based refinement, parallelized field computations, placement and adjustment of neuron models, estimation of their activity due to DBS and visualization (Fig 1). In this section, the employed modules, their functions and settings are described. It is important to note that on each step of the simulation, the platform produces metadata files. They are stored in case that the workflow was interrupted and the user wants to continue from a specified completed step.

### Setting up a preliminary discretized model

Modeling and initial discretization of the computational domain are conducted in the open-source software SALOME [8] (https://www.salome-platform.org, vers. 8.3.0) with the Netgen

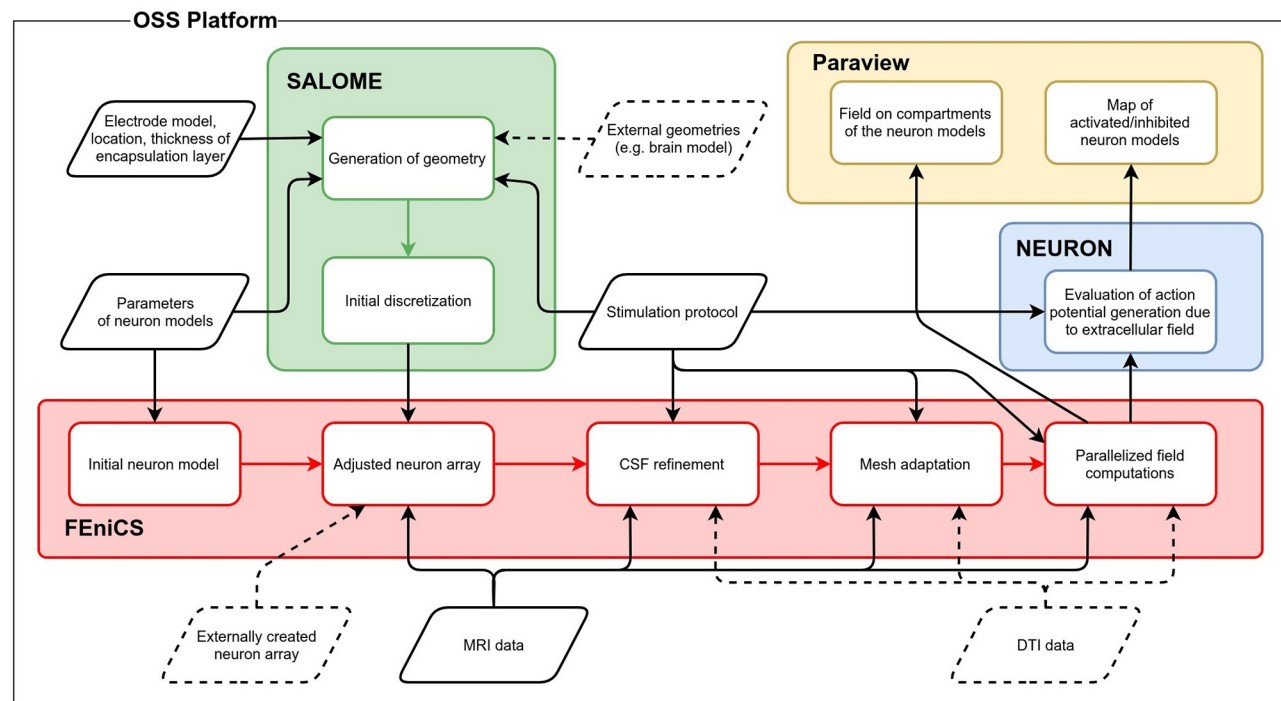

**Fig 1. Flowchart diagram of OSS-DBS with its main CAD/CAE modules.** Communication between and within the modules is conducted via various Python scripts, which form the core of the platform. Solid line parallelograms depict mandatory input data, while dashed ones are optional.

module [9]. Apart from its graphical user interface, SALOME allows to export and import
Python scripts. This provides full integration to the Python-based simulation platform and
allows to construct and discretize computational domains in an automatic mode. Due to the
fast decay of the stimulating electric field, it is valid to approximate the computational domain
by an ellipsoid with dimensions based either on magnetic resonance imaging (MRI) data sets
or manually provided values. Alternatively, a CAD model of a brain geometry can be loaded.
The developed platform supports explicit definition of geometric subdomains, and it may be
utilized to model the encapsulation layer, which is built up due to the inflammatory response
to electrode implantation. The user is offered to choose an electrode geometry from a prede-
fined collection, which contains leads for DBS in humans (e.g. Medtronic models 3387 and
3389, St Jude models 6148 and 6180) and in rodents (e.g. Microprobes SNEX-100). To create
custom electrodes in the OSS-DBS format, the user is supplied with a template file and a
manual.

The initial discretization has two aspects. Foremost, electrode contacts should be well
refined for an accurate approximation of their geometrical shapes in the model, and thus the
geometric error is estimated by the meshing algorithm. A distorted contact geometry would
lead to incorrect boundary conditions and, consequently, an incorrect electric field distribu-
tion. Contacts are usually small in comparison to the dimensions of the computational
domains, and their refinement might lead to a large number of mesh elements, especially for
tipped electrodes. The second aspect is the mesh partition: the first step of a physics-based
mesh adaptation will refine it uniformly, and in order to reduce computational costs, the pro-
cess is carried out separately in three regions (Fig 2): encapsulation layer in the vicinity of con-
tacts, region of interest (ROI), where neuron models will be placed, and rest of the tissue
(ROT). The electrode itself is removed from the computational model assuming a homoge-
neous Neumann boundary condition (BC) on the insulating surface ($\nabla \phi \cdot n = 0$). Contacts
with assigned potential/current are modeled as surfaces where Dirichlet BC is applied. Inactive

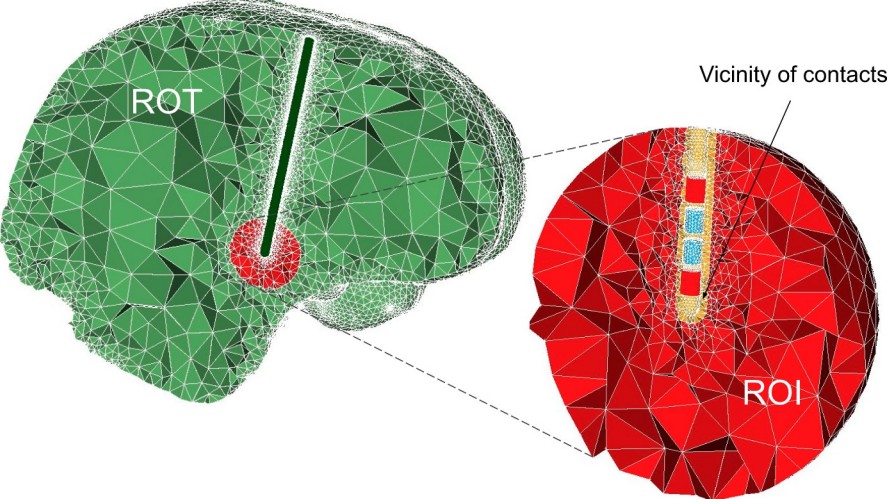

**Fig 2. Example of an initial human brain discretization in OSS-DBS.** The model is divided into the region of
interest (ROI), the vicinity of contacts and the rest of the tissue (ROT). All three have their own submeshes with
different element size requirements. Additionally, submeshes are defined for the electrode contacts and the
encapsulation layer away from them. Controlled surface refinement of the contacts leads to a highly dense submesh
around the electrode. Here, the active contacts are depicted in red, while the floating conductors are presented as blue
cylinders.

contacts are modeled as floating conductors with the virtual permittivity method [10] (relative permittivity is set to $10^9$).

## Placement and adjustment of neuron models

In this paper, neuron models are not of primary interest. However, in order to estimate the influence of the model's sophistication, activation of mammalian myelinated axons is investigated. The employed computational model is based on a double cable structure with explicit modeling of nodes of Ranvier and internodal segments [11]. The activation arises from the membrane polarization driven by the extracellular electric field. In [7], authors have created Python libraries for axons with different fiber diameters and prepared routines for computations in the NEURON environment [12], in which the axon model is defined. These scripts were adopted for the use in OSS-DBS. The user can choose the axonal fiber diameter and the number of nodes of Ranvier, while internodal segments will be generated automatically. Alternatively, the user can load their own axonal geometries, represented as a sequence of points, at which the electric potential will be probed. From the constructed neuron model, the platform can build structured arrays (Fig 3A) or deploy models with coordinates and angle vectors provided manually. It is also possible to import realistically placed axonal populations (Fig 3B) to investigate the pathway activation due to DBS. Afterwards, the adjustment algorithm will delete all those models whose segments are located outside of the computational domain, inside the encapsulation layer, floating conductors or in a specified tissue (e.g. cerebrospinal fluid (CSF)), as these models are considered unrealistic or damaged. The spatial extent of the specified tissue is obtained from the MRI data set. Midpoints on the segments of the remaining neuron models will be used for mesh adaptation and estimation of the axonal activation.

## Physics of volume conductor model

In the simulation platform, the user defines the stimulation signal choosing its shape (rectangular, centered triangular, ramps), frequency, amplitude, pulse width and phase. The

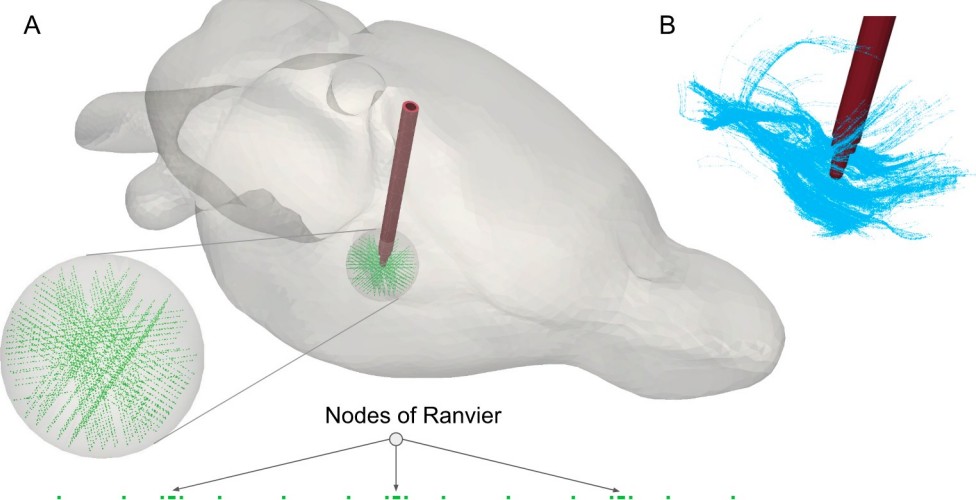

**Fig 3. Representation of axon arrays in OSS-DBS.** In the volume conductor model, an axon is defined as a sequence of points (nodes of Ranvier and internodal segments), with spacing dependent on axonal morphology. If at least one segment lies in the encapsulation layer (in red), cerebrospinal fluid or outside of the computational domain, the axon is excluded. (A) Ordered axon array employed for VTA estimations. (B) Realistically placed axons passing in the vicinity of the subthalamic nucleus derived from the fiber tractography of the rat brain [13].

generated field is computed in frequency domain using the Fourier Finite Element Method (FFEM) [6]. In DBS, electric potential is estimated by solving the EQS formulation of Maxwell's equations [14]:

$$\nabla \cdot ((\sigma(\mathbf{r}, \omega) + j\omega\varepsilon(\mathbf{r}, \omega))\nabla\underline{\phi}(\mathbf{r})) = 0. \tag{1}$$

The EQS approximation is valid for electric fields of relatively low frequencies in the absence of magnetic induction [15]. Here, $\underline{\phi}$ is the complex electric potential, $\omega$ denotes the angular frequency and j is the imaginary unit, $\varepsilon$ and $\sigma$ are the permittivity and conductivity of the material, respectively. OSS-DBS solves the EQS problem using the open-source software FEniCS [16] (https://www.fenicsproject.org, vers. 2017.2.0), which contains programming and mathematical tools for solving partial differential equations with the FEM. To account for complex numbers, the problem is formulated on a mixed function space, comprised of Lagrange finite elements both for the real and the imaginary parts. Dirichlet BC are defined as potentials on the electrode contacts, with the imaginary part set to zero. Nowadays, current-controlled stimulation draws attention due to its higher persistence against electrical double layer effects [6]. The platform supports this stimulation mode by means of field scaling, assuming linear properties of the FEM solution. The scaling factor is the current through the active contact:

$$\underline{J}_n(\mathbf{r}) = \oiint\limits_{S_{\text{contact}}} (\sigma(\mathbf{r}, \omega) + j\omega\varepsilon(\mathbf{r}, \omega))\underline{E}_n(\mathbf{r})\mathrm{d}S_{\text{contact}}. \tag{2}$$

Dividing the electric potential by this value yields the potential distribution for 1 A. The linearity of the system also allows to estimate the effect of different stimulation amplitudes without recalculation of Eq (2): the potential on the neuron compartments is simply scaled. However, this approach has a limitation for the multicontact case: it can be applied only for the voltage-controlled stimulation and only if voltages on all contacts are equally scaled.

In case of a multicontact current-controlled stimulation, the system is solved for each contact with assigned current and the ground, while the rest of the contacts are set to floating potentials. The Dirichlet BC for contact $k$ is then defined as

$$\underline{V}_{k/Jk} = \frac{V_{Jk}}{\underline{I}_{n,k}}J_{Jk} + \sum_{m \neq k} \underline{V}_{k/Jm}\frac{J_{Jm}}{\underline{I}_{n,m}}, \tag{3}$$

where $\underline{V}_{k/Jk}$ is the scaled voltage on the contact to obtain assigned current $J_{Jk}$; $V_{Jk}$ is the voltage set to the value of prescribed current and $\underline{I}_{n,k}$ is the calculated current on the contact for this voltage. $\underline{V}_{k/Jm}$ is the floating potential computed on contact $k$ when contact $m$ is used as Dirichlet BC. The first term in Eq (3) is Ohm's law to find the voltage with a fixed impedance (the fraction), while the second ensures that no additional current will be delivered or extracted through the contact when other contacts are active. Note that assigned values are always real, while computed complex.

The frequency range in Eq (2) is defined by the Fourier transform of the stimulation signal, and the dispersive nature of brain tissue is taken into account using a 4-term Cole-Cole model with parameters from [17] (Fig 4). In some research, the capacitive term is neglected assuming purely conductive brain tissue, i.e. the quasistatic (QS) case. This formulation significantly reduces the computational cost, but its application for a conventional DBS signal may lead to considerable errors in evaluating neural activation, which is highly sensitive to the field distribution [6]. Fig 4B demonstrates the magnitude of the capacitive term over the frequency range of the Fourier transformed DBS pulse. Nevertheless, the QS formulation is supported by the

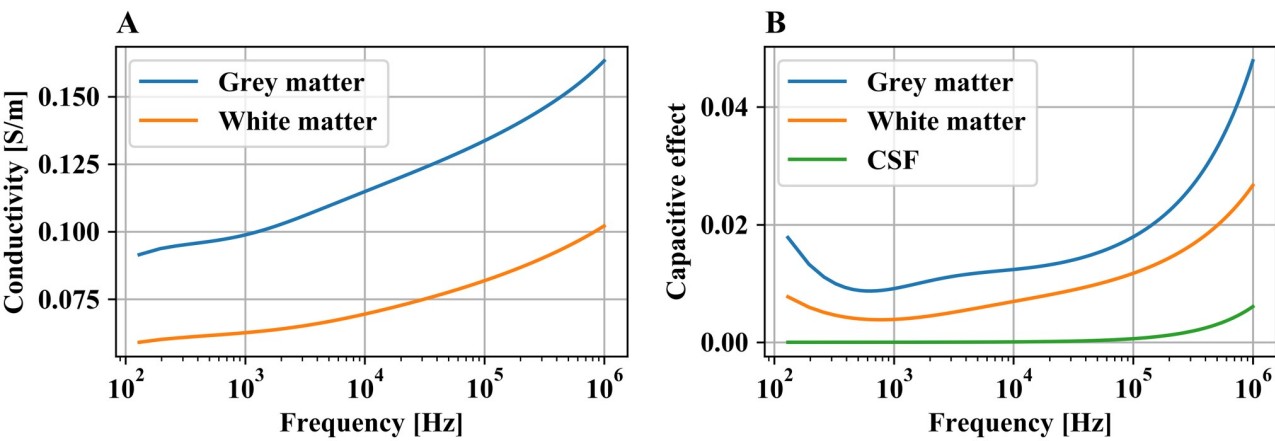

**Fig 4. Dielectric properties of brain tissue in the frequency spectrum of a 60 $\mu$s rectangular pulse with a repetition rate of 130 Hz.** (A) Conductivity of grey and white matter. (B) Capacitive effect of brain tissue ($\omega\varepsilon$).

platform in case the user expects or observes a low effect of tissue capacitance on the field solution.

The presence of $\sigma$ and $\varepsilon$ in Eq (2) and their varying values for brain tissue motivate to consider heterogeneity as it influences the current distribution [18]. OSS-DBS supports the incorporation of segmented MRI data (in NIfTI or .txt grid format) by mapping it onto the mesh (Fig 5). The accuracy of the mapping depends on the level of discretization, introducing additional criteria for the adaptive refinement. Furthermore, the targets of DBS reside near fiber tracts with highly anisotropic properties, and thus considerable effect on the electric field

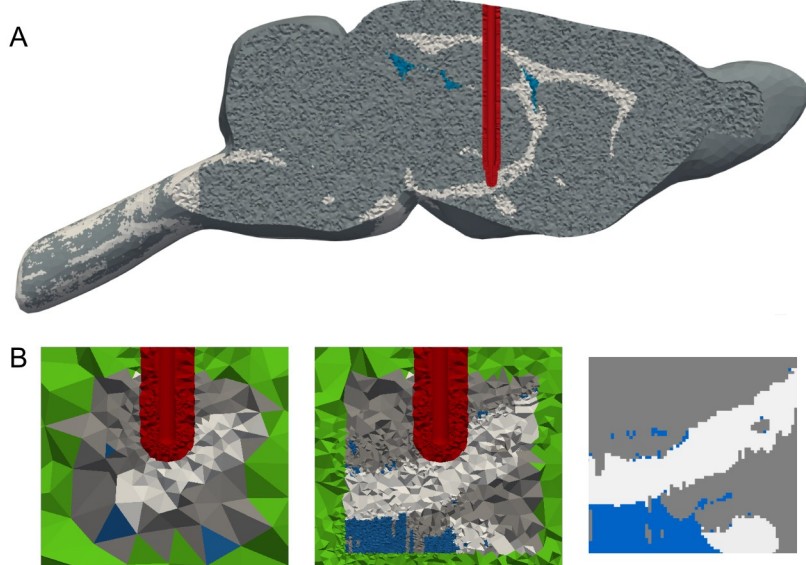

**Fig 5. Distribution of the rat brain tissue [13], mapped onto a tetrahedral mesh using OSS-DBS.** Grey and white matter are depicted with the corresponding colors, CSF is shown in blue and the encapsulation layer in red. The electrode lead is considered to be a perfect conductor and subtracted from the computational domain. (A) Tissue map for the whole brain. (B) Local mapping of the tissue before (left) and after (center) CSF refinement. The unmapped cells are depicted in green. For comparison, the segmented MRI data are shown on the right.

distribution is imposed [19]. In the simulation platform, anisotropy is implemented analogously to [20], using conductivity tensors, obtained from voxelized Diffusion Tensor Imaging (DTI):

$$\nabla \cdot \left( \left( \begin{bmatrix} D_{xx} & D_{xy} & D_{xz} \\ D_{xy} & D_{yy} & D_{yz} \\ D_{xz} & D_{yz} & D_{zz} \end{bmatrix} \sigma(\mathbf{r}, \omega) + j\omega \begin{bmatrix} 1 & 0 & 0 \\ 0 & 1 & 0 \\ 0 & 0 & 1 \end{bmatrix} \varepsilon(\mathbf{r}, \omega) \right) \nabla \underline{\phi}(\mathbf{r}) \right) = 0. \tag{4}$$

Components $D_{ij}$ of the conductivity matrices are weighting factors, and for tissue with weak anisotropy the diagonal values are close to 1, while others to 0. Anisotropy in the encapsulation layer and the floating conductors, as well as for permittivity is not considered. In case of fragmentary MRI/DTI data, local mapping is available (Fig 5B).

## Mesh adaptation

As mentioned before, the convergence of the solution on neuron model compartments is the criterion for mesh adaptation. The adaptation is carried out in two steps. At first, the algorithm evaluates the effect of CSF refinement on the pointwise solution for the magnitude of the electric potential, represented by $\phi_{aim}$. CSF is 10-20 times more conductive than grey and white matter, and therefore the accurate mapping of CSF space is crucial. The adaptive algorithm starts by refining all cells that contain CSF voxels (Fig 5B) and reside in the vicinity of the deployed neuron models until the cells do not exceed a certain size (e.g. MRI voxel size). Next, $\phi_{aim}$ is computed on the obtained mesh. The CSF refinement trigger criterion is defined as

$$\left\| \frac{\phi_{aim} - \phi_k}{V_{drop}} \right\|_\infty > \Theta_{CSF}, \tag{5}$$

where $\phi_k$ is the solution in the $k$-th refinement iteration, $V_{drop}$ is the magnitude of the voltage drop across the tissue, and $\Theta_{CSF}$ is the deviation threshold for the electric potential on the neuron compartments during CSF refinement. The zero iteration corresponds to the initial mesh and the subsequent iterations have descending cell size for CSF voxels.

In the second step, the refinement is carried out in the three previously defined regions. Firstly, a submesh is refined uniformly and $\phi_{new}$ is computed. The trigger criterion for adaptive refinement is

$$\left\| \frac{\phi_{new} - \phi_{old}}{V_{drop}} \right\|_\infty > \Theta_\phi, \tag{6}$$

where $\Theta_\phi$ is the deviation threshold during adaptive refinement. If Criterion 6 is not fulfilled, the algorithm refines cells in the subdomain on the initial mesh, where the change of the solution for the electric field $\underline{E}$ is above the relative field deviation threshold $\Theta_E$

$$\frac{\|\underline{E}_{new} - \underline{E}_{old}\|_2}{\|\underline{E}_{new}\|_2} > \Theta_E, \tag{7}$$

computes $\phi_{new}$ and checks Criterion 6 again. Further refinement will be conducted on the mesh from the preceding iteration. If the required convergence is achieved, the adapted region will be tested again with a uniform refinement to avoid the local convergence effect. The scheme is concisely described in Fig 6. As it was mentioned before, the trigger criteria are evaluated on the compartments of the neuron models when the meshes are compared

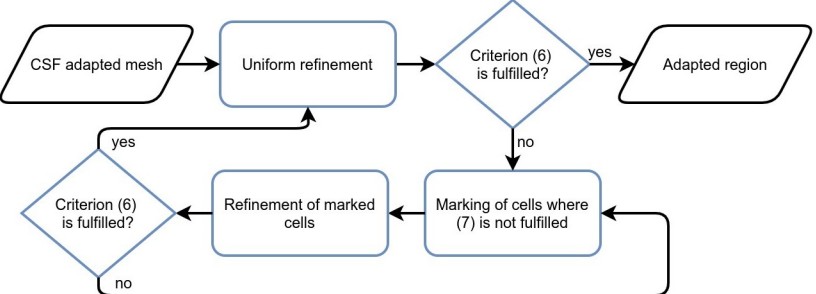

**Fig 6. Flowchart of adaptive mesh refinement.**

qualitatively. However, to locate cells with a poor convergence, Criterion 7 is estimated on midpoints of cells in the region.

The current-controlled mode requires an additional evaluation of a current convergence criterion. The criterion is based on the change of currents integrated over mesh interfaces of electrode contacts. If the change due to refinement exceeds a prescribed threshold, the refinement algorithm looks for cells where

$$\frac{\sum_{i=1}^{n} \left( \iiint_{cell} |\underline{\mathbf{J}}(\mathbf{r})| d\Omega_i \right) - \iiint_{cell} |\underline{\mathbf{J}}(\mathbf{r})| d\Omega}{\sum_{i=1}^{n} \iiint_{cell} |\underline{\mathbf{J}}(\mathbf{r})| d\Omega_i} > \Theta_{\mathbf{J}}, \tag{8}$$

Here $\underline{\mathbf{J}}(\mathbf{r})$ is the complex current, *Cell* refers to the cell in the initial mesh, $n$ is the number of *cells* that comprise *Cell* in the refined mesh and $\Theta_{\mathbf{J}}$ is the relative deviation threshold for the current. This condition will not be evaluated in the cells with small currents as their refinement will not influence the solution. It is important to note that the current is not dependent on the potential itself, but its spatial derivative. Therefore, in order to ensure a non-constant gradient over a cell, the platform employs at least the second order basis functions for the evaluation of the electric potential.

## Impedance of the computational model

An estimation of the current on the contact is also required for simulations with the electrode-tissue interface, often denoted as electrical double layer, formed due to metal-electrolyte interaction. Effects on the interface are described by two mechanisms: the non-Faradaic (pseudocapacitive) flow and the Faradaic processes, where actual charge transfer occurs. Contribution of the former to the overall impedance can be described by a constant phase element (CPE) [21]:

$$Z_{\mathrm{CPE}} = \frac{K_{\mathrm{S}}}{(\mathrm{j}\omega)^{\alpha}}, \tag{9}$$

where $K_{\mathrm{S}}$ is the scaling factor of the CPE and $\alpha$ describes the relation between resistance and reactance. $K_{\mathrm{S}}$ and $\alpha$ depend on frequency, material, geometry and surface roughness of the contact and remain constant only at low currents. In computational models for DBS in humans, only the non-Faradaic processes are usually assumed as the electrode contacts are made of platinum-iridium alloy and considered to be polarizable. The interface is assumed to be in series with the tissue, and thus the field distribution during current-controlled stimulation is considered to be non-affected by the interface impedance and its fluctuations.

In contrast to this, during a voltage-controlled stimulation the electrode-tissue interface will affect the electric field in brain tissue. In the platform, the interface is not simulated explicitly due to computational costs. Instead, the assigned potentials on two active contacts are modified by the voltage drops over the CPEs:

$$V_{\text{CPE1}} = \frac{V_{\text{source}}}{Z_{\text{tissue}} + Z_{\text{CPE1}} + Z_{\text{CPE2}}} Z_{\text{CPE1}}, \tag{10}$$

where $Z_{\text{tissue}}$ is defined as $V_{\text{source}}/J_{\text{source}}$, and $J_{\text{source}}$ is obtained from a simulation without the electrode-tissue interface using Eq (2). Then, the Eq (1) is solved again for the new boundary conditions. It is also important to note that the accuracy of MRI/DTI mapping can be evaluated by convergence analysis of $J_{\text{source}}$ as it depends on tissue conductivity and permittivity. The largest deviations are normally observed during refinement in the vicinity of the active contacts, where the electric field is the highest. The error is amplified if the local mesh resolution is too low to map the MRI/DTI data correctly. In OSS-DBS, the mapping accuracy close to the contacts is ensured by the aforementioned adaptive mesh refinement, and especially, the current convergence criterion.

## Parallel computations in OSS-DBS

As previously mentioned, the application of a rectangular DBS pulse necessitates solving EQS for a multitude of sine-wave stimuli. Their individual contributions are calculated using a Fourier transformation of the DBS signal. The transformation contains an infinite number of components, and a frequency range of up to 1 MHz should be considered in order to achieve a sufficient approximation of the conventional DBS signal. However, the solutions for different frequencies are decoupled, and therefore the computations can be efficiently parallelized. The same is valid for the inverse Fourier transformation (IFT) of the electric potential, which is applied to obtain the time-dependent solution on the neuron compartments. In this case, parallelization becomes especially meaningful if a large amount of neurons is deployed. Furthermore, assuming a dominating effect of the DBS input, the elicitation of action potential in neuron models is also simulated in parallel. The mechanism is implemented in the platform using the Python multiprocessing library and can be utilized on ordinary multicore workstations. For the aforementioned problems, the parallelization works with a strong scalability, and for the best performance it is recommended to employ all available physical CPUs taking into account possible memory limitations. To speed up FEM computations during adaptive mesh refinement, OSS-DBS supports MPI for FEniCS. However, this option should be used with caution as the solution convergence and the performance scalability depend on the conditioning of the stiffness matrix and the applied FEM solver. A comparison of the parallelization performance for the simulation problems is presented in S1 Fig.

Additionally, the platform offers two types of frequency spectrum truncation methods. With the first, the user manually chooses the number of frequencies for calculations, and the frequencies will be either picked sequentially or basing on the magnitude of corresponding sine-wave stimuli in the Fourier transformation. The second method requests a frequency, above which the calculations will be conducted using octave bands. Choice of a method depends on the complexity of the volume conductor model and number of neuron compartments. Please refer to [22] for more details.

## User-platform interaction

To simplify the setup of DBS studies and the interaction with OSS-DBS, a graphical user interface (GUI) was developed, where the user defines inputs and simulation parameters, which

will be converted to a Python dictionary, see S2 Fig. For a quick start, a collection of predefined simulation setups is provided with a description that explains users how to assess the generated results.

In each step of the simulation, the platform produces output files for intermediate analysis. They can be visualized using Paraview [23] (https://www.paraview.org). If the platform is installed on a server, and the user works via remote access, the GUI of Paraview might be inconvenient for displaying large data sets. In this case, the user is offered a gallery of screenshots, generated without launching the interface by a prepared collection of Paraview scripts. When a simulation is finished, the platform will display the deployed axon models and highlight the activated ones. Additionally to the direct estimation of axonal activation, the platform can compute approximations, which are based on thresholds for the electric field or its derivative.

### Docker image for OSS-DBS

Multiple CAD/CAE tools and Python packages are utilized in OSS-DBS. Proper installation of all these open-source software products might not be a simple task, especially for users who are not confident with Linux systems. For this reason, and to port the platform to other operating systems, it was decided to create a docker image (https://www.docker.com/). The main idea is to encapsulate software services into a so-called container that can be easily shared and run on different hosts, e.g. a server in the clinical research environment or a powerful workstation. Compared to full-stack virtual machines like VMware or KVM, container solutions provide comparable or better performance [24]. To ensure runnability of the container, automatic tests were implemented using Continuous Integration inside the Github Repository.

## Results

The following results can be reproduced with input dictionaries and binary files available in the project's repository https://github.com/SFB-ELAINE/OSS-DBS/tree/master/OSS_platform/Example_files/Publication%20results.

### Benchmarking

To evaluate the accuracy of the volume conductor models generated by the platform, analogous computations in COMSOL Multiphysics 5.5 were conducted. COMSOL is a commercial FEM software and it is conventionally used for electric field calculations in DBS *in silico* studies. It should be noted that the performance of the software is not compared here, as it is highly dependent on the problem specification. Hardware specification and simulation details are provided in S1 Appendix.

Two setups were tested: voltage-controlled stimulation in a human brain tissue with an ordered axon array (as for VTA estimations) and current-controlled mode in a rat brain tissue with realistically placed axons (as for estimations of pathway activation). In both cases, the computational domain was truncated to spheres 32 and 10 mm in diameter, respectively, centered on the subthalamic nucleus. The comparison was conducted for 520, 5,200, 52,000, and 520,000 Hz sine waves to estimate the modeling accuracy over the frequency spectrum. The dielectric properties were chosen according to [17] and presented for 520 Hz in Table 1.

**Table 1. Dielectric properties of brain tissue at 520 Hz [17].**

| Tissue Type | Grey Matter | White Matter | CSF | Encap. layer |
|---|---|---|---|---|
| $\sigma$ [S/m] | 0.0964 | 0.0616 | 2.0 | 0.048228 |
| $\varepsilon_r \cdot 10^4$ | 30.407 | 13.752 | 0.0109 | 30.407 |

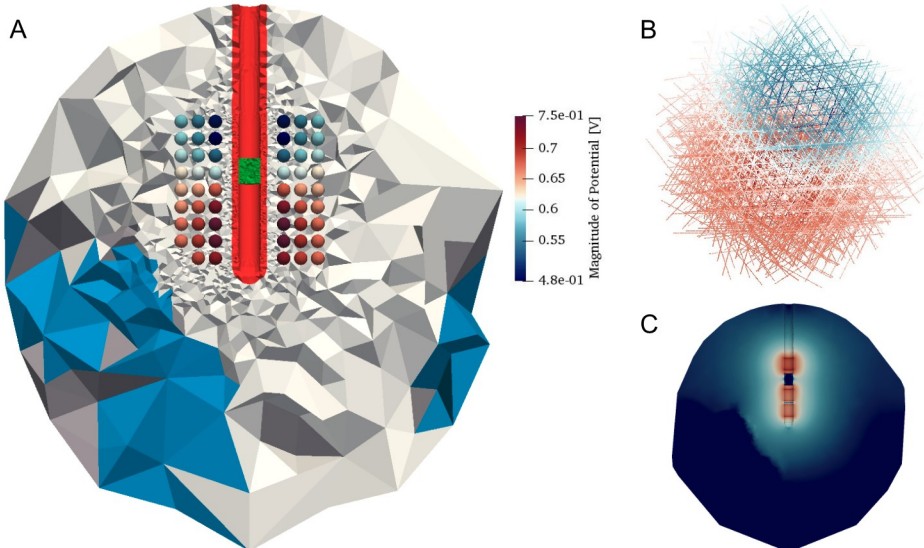

**Fig 7. Results of voltage-controlled stimulation in the human brain tissue.** (A) Distribution of the brain tissue and a floating conductor (in green) on the adaptively refined mesh and the electric potential distribution on transversally aligned axons shown in one plane. Note that one of the axons was subtracted due to unrealistic placement. (B) Distribution of the electric potential magnitude on the ordered axon array. (C) Distribution of the electric field magnitude in the computational domain (log scale). The shape is evidently distorted by highly conductive CSF.

For the voltage-controlled setup, the digital SRI24 multi-channel brain atlas [25] was used to map grey, white matter and CSF onto the computational domain (Fig 7A) to take into account the heterogeneity of the tissue. Medtronic model 3389 (Medtronic Inc., Minneapolis, US) was chosen as the stimulating electrode, with the two lowest contacts set to 1.0 V and the upper contact defined as the ground, while the upper center contact was set to a floating potential. To obtain the electric potential distribution using the EQS formulation in COMSOL, the "Electric Currents" module from AC/DC physics was employed. Although the models had the same parameter setup and initial meshed geometry, the discretization was intentionally modified to estimate the performance of the adaptive refinement algorithm in OSS-DBS. An extremely fine mesh with the overall number of 35,943,642 elements, ineligible for multiple computations, was created in COMSOL as the benchmark. In the platform, an adapted mesh for 520 Hz sine wave signal consisted only of 586,598 tetrahedrons. For both cases, the second order FEM basis functions were employed.

The accuracy of the computations was estimated by the discrepancy of electric potential magnitude on the neuron compartments (Fig 8B). The results showed that the maximum difference lied below 0.01 V (i.e. less than 1% of the total voltage drop), while the average difference amounted to 0.002 V. Computed electric field distribution (Fig 7C) revealed a prominent effect of the CSF on the EQS solution.

A volume conductor model for the current-controlled setup (Fig 8A) was created using segmented Waxholm Space atlas of the Sprague Dawley rat brain [13] and bipolar SNEX-100 electrode (MicroProbe Inc., MD, USA). The core contact was assigned 0.1 mA and the outer contact set to ground. Axonal compartments were allocated on fiber tracts [13] passing in the vicinity of the subthalamic nucleus (Fig 8B). The COMSOL benchmark model was built with 3,809,230 elements and employed third order basis functions, while the mesh adapted in the platform at 520 Hz consisted of 735,484 tetrahedrons with the application of the second order functions.

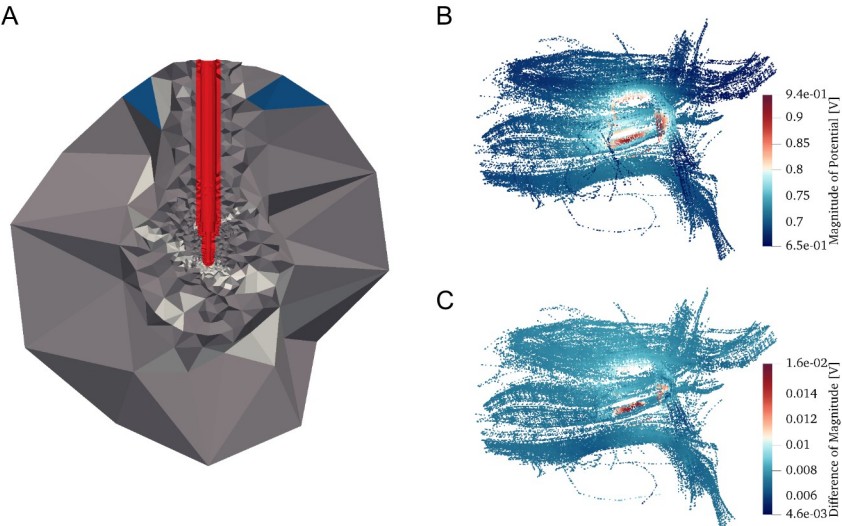

**Fig 8. Results of current-controlled stimulation in the rat brain tissue.** (A) Distribution of the brain tissue on the adaptively refined mesh. Note the coarse discretization on the periphery of the computational domain which is sufficient due to the fast decay of the electric field away from the electrode contacts. (B) Distribution of the electric potential magnitude on the realistically placed axons in the vicinity of the STN. (C) Electric potential magnitude difference on the axonal compartments computed with the OSS-DBS and COMSOL models.

For this case, the maximum difference on the neuron compartments reached 0.017 V (0.7% with the total voltage drop of 2.37 V), while the average difference amounted to 0.0077 V (Fig 8C). Furthermore, in both cases, the compartments with relatively high discrepancies were found to be located in the close vicinity of the contacts. Usually, such axons are considered to be inevitably activated due to a high local electric field. The simulations for the 5,200, 52,000, and 520,000 Hz sine-wave stimuli yielded nearly the same deviations.

## Evaluation of physics in volume conductor model

In the platform, different physics for the VCM design and field simulation were implemented. It is of particular interest to assess qualitatively the effects of the QS formulation and CPE on the electric potential distribution, as these physics affect the impedance of the computational model. It was previously reported that for current-controlled stimulations the capacitive term in Eq (1) leads to the rise of the voltage during the DBS pulse [26], while CPE has an opposite effect during voltage-controlled stimulations [6]. The corresponding computational models were designed using the platform, and the results demonstrated the expected behavior (Fig 9), validating the implementation of the physical model.

## Discussion

The prediction and enhancement of DBS outcome relies on simulation results, whose accuracy is highly dependent on the VCM. Its development is a complex procedure that includes geometry design, its discretization and mesh convergence analysis; incorporation of heterogeneous and anisotropic dielectric properties, consideration of electrochemical effects on electrode contacts and application of appropriate mathematical apparatus for electric field computations. At the same time, the VCM should be as computationally efficient as possible due to the multiple calculations required for FFEM. Moreover, high efficiency is crucial for the solution of optimization problems, which also necessitate automated generation of different VCMs.

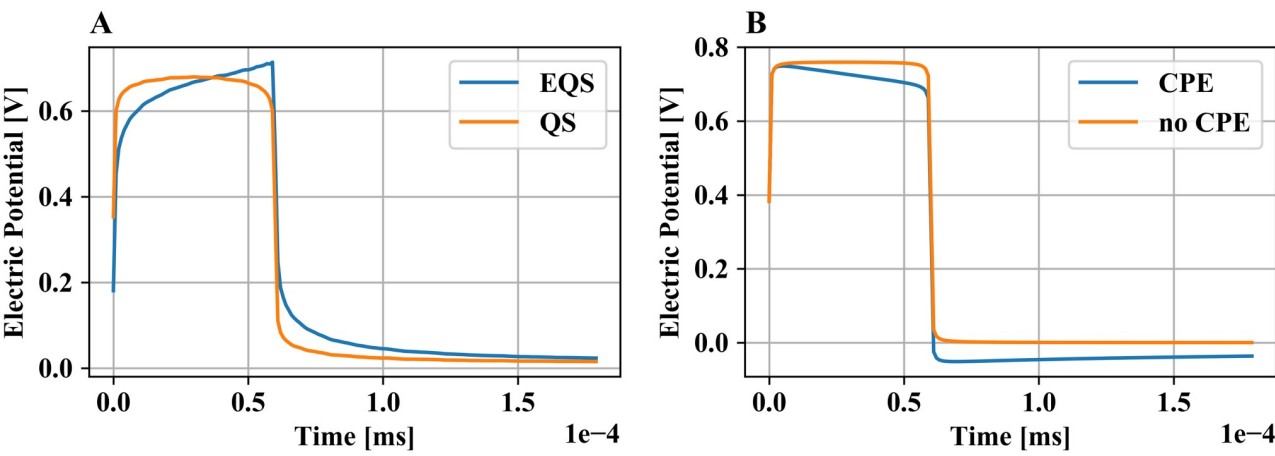

**Fig 9. Electric potential distribution on a single axonal compartment during DBS with different computational models.** (A) Current-controlled stimulation in the rat model, computed with the EQS and the QS formulations, both without CPE. The difference arises from the capacitive charging in the former formulation. (B) Voltage-controlled stimulation with and without CPE in human model for the EQS formulation. The shape of the potential is affected by the high impedance of the electrical double layer at low frequencies.

In this paper, we presented a simulation platform that meets the described requirements while employing strictly open-source software. Various simulation tools for DBS have already been developed [7, 27–29], but to the best of our knowledge, either the computational model was comparatively simplified, or the design was conducted manually with a low feasibility of optimization, or commercial software was employed. Our work has a structural resemblance with [7], from which the workflow was partially adapted. In contrast to [7], the workflow is automated in the platform, significantly reducing the prototyping effort. For the same reason, an adaptive mesh refinement algorithm was introduced. To facilitate the assignment of dielectric properties, the platform supports direct mapping of segmented MRI data. Moreover, the anisotropy of tissue is taken into account using tensor-weighted EQS. Its solution is parallelized in frequency domain to reduce the total computation time. Additionally, such aspects as voltage drop across CPE, frequency spectrum truncation and neuron model adjustment are implemented. A recent study on advanced volume conductor modeling in FEniCS [30] employs similar approaches for solving different formulations of Eq (1) to calculate the electric field distribution including implementation of FFEM. However, OSS-DBS is more focused on comprehensive and automated computations for deep brain stimulation, thus providing a wider functionality that includes: automated geometry generation, direct mapping of dielectric properties from medical imaging data, adaptive mesh refinement, truncation methods for FFEM, integration of neuron models, etc.

The accuracy of OSS-DBS was tested on relevant DBS setups using the commercial software COMSOL Multiphysics® and the results were found to be in a close agreement. For the presented examples of a voltage-controlled stimulation in the human brain tissue and a current-controlled stimulation in the rat brain tissue, the discrepancy in the electric potential on the neuron compartments did not exceed 1%, while the computational complexity was significantly lower for the OSS model. A good agreement was also observed in other simulation setups and for different physical models.

## Application of OSS-DBS

The primary aim of the platform presented in the paper is to simplify simulations of DBS while preserving their accuracy. Our tool can be used for a quick assessment of stimulation

protocols or by computational scientists for a complex optimization and UQ analysis. The latter is of a special interest for the DBS community due to uncertainties in brain tissue properties and electrode placement [31–34]. It is important to emphasize that the platform does not discriminate between targets of DBS or species due to a high parametrization of the algorithms for VCM modeling.

Furthermore, the modules of the platform can be applied in computational models for other electrically active implants. For example, MRI/DTI mapping offers a viable alternative to 3D modeling of the regions with high heterogeneity and/or anisotropy; the introduced adaptive mesh refinement routines are of interest for costly simulations with pointwise defined outputs. For problems dealing with neuron populations, the generation and adjustment algorithm, outlined in this paper, might be of use.

## Future development and limitations

In order to conduct a simulation in OSS-DBS, the user has to setup their study using the GUI. Most of its entries, such as implantation site or number of CPUs, can be given directly. However, MRI/DTI data sets, CAD models of brains or electrode leads should be provided in external files. Furthermore, information about entries like CPE parameters or thickness of the encapsulation layer might not be at hand. At the moment, a limited library of these data is offered, and in the future it is planned to expand the collection according to requirements of users.

The computational speed remains an issue, especially in context of iterative optimization and UQ studies. To tackle the problem, a message passing interface was employed to distribute solution of the FEM problem over CPUs. Currently, we are developing a two-level parallelization for computer clusters: not only calculations in frequency domain will be carried out in parallel, but also the solution of linear systems of equations.

In the upcoming versions, special focus will be put on neurons and their morphology. Axon models will be connected with corresponding somatic compartments and dendritic trees for evaluation of a coupled response to the DBS signal. For the cases where explicit simulations of neurons are not of primary interest, an approximate approach, such as driving force method [35] will be added.

Currently, new mechanisms for dielectric properties mapping are being tested. In our customized studies, heterogeneity is additionally imposed by imported brain/head structures (e.g. ventricles or skull), which allows to reduce significantly the mesh size. Application of DTI data for anisotropy can be troublesome due to different tensor normalization techniques or data unavailability. Therefore a new approach was developed where FEM elements are assigned conductivity tensors based on the direction and amount of fiber tracts passing in the vicinity. Both mechanisms will be added in the upcoming version of OSS-DBS.

## Conclusion

In an ageing society, an increasing number of patients suffer from diseases for which DBS presents a successful therapy. Both for research and for patient-individual therapy planning, an easy-to-use, non-commercial simulation platform is desirable that allows for a highly automated simulation pipeline starting from the patient-specific data up to reliable statements about the effective stimulation. Here we present such an open-source simulation platform that generates computational models for DBS in an automated manner. The computational models take into account tissue heterogeneity and anisotropy, non-Faradaic processes and different axonal morphologies. It supports different stimulation modes and pulses as well as different VTA estimates. The platform uses adaptive mesh refinement algorithms, parallel processing,

and frequency truncation methods to ensure a high efficiency of the computations. Due to its automated performance, the platform is well suited for *in silico* research on optimization or UQ problems in DBS. VCMs, generated by the platform for two atlas-based data sets for the human and rat brain, were validated against commercial software and literature results.

## Supporting information

**S1 Fig. Parallelized performance of OSS-DBS for different simulation steps of a test study (available in the repository as Example dict quick test.py), which was conducted in a Docker container.** The bars show computational time as % to the single CPU run.
(TIF)

**S2 Fig. Graphical user interface of OSS-DBS to set up simulations.**
(TIF)

**S1 Appendix. The benchmark simulations were conducted in a Docker container on Intel Xeon(R) Gold 6136 CPU @ 3.00 GHz x 48 machine with 376.6 GB of memory, with disabled MPI, i.e. only on one core, and up to 128 GB used by OSS-DBS.** The adaptive mesh refinement for the human and the rat brain tissue took 1 hour 46 minutes and 3 hours 54 minutes, respectively.
(TXT)

## Acknowledgments

The authors wish to thank Andrea Andree for her advising and assistance with the MRI/DTI data processing.

## Author Contributions

**Conceptualization:** Ursula van Rienen.

**Funding acquisition:** Ursula van Rienen.

**Investigation:** Konstantin Butenko.

**Methodology:** Konstantin Butenko, Christian Bahls, Max Schröder.

**Project administration:** Ursula van Rienen.

**Resources:** Ursula van Rienen.

**Software:** Konstantin Butenko, Max Schröder.

**Supervision:** Christian Bahls, Rüdiger Köhling, Ursula van Rienen.

**Validation:** Konstantin Butenko.

**Visualization:** Konstantin Butenko.

**Writing – original draft:** Konstantin Butenko, Rüdiger Köhling.

**Writing – review & editing:** Christian Bahls, Rüdiger Köhling, Ursula van Rienen.

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
