## [Decision Letter · Decision Letter 0]

6 May 2020

Dear Mr. Butenko,

Thank you very much for submitting your manuscript "OSS-DBS: Open-Source Simulation platform for Deep Brain Stimulation with automated volume conductor model design and advanced field modeling." for consideration at PLOS Computational Biology. As with all papers reviewed by the journal, your manuscript was reviewed by members of the editorial board and by several independent reviewers. The reviewers appreciated the attention to an important topic. Based on the reviews, we are likely to accept this manuscript for publication, providing that you modify the manuscript according to the review recommendations.

Please make sure that the code and material necessary to reproduce and extend the results is correctly accessible on a third party repository (this will be checked prior to acceptance).

Also as you anticipated, make sure that you address the paper on a similar topic recently published which you brought our attention to.

Sincerely,

Daniele Marinazzo

Deputy Editor

PLOS Computational Biology

Daniele Marinazzo

Deputy Editor

PLOS Computational Biology

[LINK]

Reviewer's Responses to Questions

**Comments to the Authors:**

Reviewer #1: In their manuscript entitled “OSS-DBS: Open-Source Simulation platform for Deep Brain Stimulation with automated volume conductor model design and advanced field modeling.” – Butenko et al. propose a novel open source toolbox to estimate electrical models of deep brain stimulation electrodes surgically implanted in rodents and humans.

Already, the van Rienen group is one of the leading groups working on precise bioelectrical models and the group has shown similar outstanding work in the past. Making their pipeline available in form of an easy-to-install software is highly relevant to the field of DBS since set up of commercial tools requires costly & time-consuming infrastructure and is not feasible in every lab studying the effects of DBS.

Hence, I do believe this will be a major addition to the field and will more broadly enable studying the complex effects of DBS onto the brain. Currently, there also is no openly available alternative that is capable of what OSS-DBS is able to do.

The manuscript is very well written, methods are accurate & figures are appropriate. I have only minor things to add and one suggestion that could be amended if possible without major additional work.

Namely, one open source alternative that is broadly used in the more clinically oriented field (Lead-DBS, www.lead-dbs.org, also cited by the authors) comprises of a much more limited FEM-based VTA model. Thus, a direct exemplary comparison between results from Lead-DBS (say in a single patient) and OSS-DBS could be interesting to the medical community. Are results similar? If they are different, what reasons may apply? Which settings in the larger parameter space of OSS-DBS would lead to similar solutions as in Lead-DBS?

Again, authors should see this as a suggestion only.

Minor:

- Are the insulating parts of the electrodes explicitly modeled? I do think those would have an effect on the shape of the stimulation volume.

- I think dystonias (plural) in the introduction is a bit uncommon (singular dystonia would be more appropriate here).

- St. Jude Medical was bought by Abbott, so maybe specifying electrode types with the novel name is more appropriate.

Reviewer #2: In this manuscript, the authors introduced a python-based platform (OSS-DBS) to control workflow for simulating deep brain stimulation, which provides an efficient and more realistic volume conducting model with the easy-to-use automated user interface.

The proposed workflow utilized open-source software and libraries to facilitate accessibility and reproducibility of computational simulation on both clinical and research applications of DBS.

The Authors also performed the benchmarks the accuracy of their proposing model by comparing with commercial software (COMSOL). Using this result, they demonstrated the platform shows high efficiency and accuracy in estimating the volume of tissue activation (VTA) for both human and rat brain by utilizing their mash adaptation refinement algorithm.

However, the lack of presenting any example regarding the user-interface is one major downside of this manuscript. So this reviewer would like to recommend to the authors to provide usage examples (with some screenshot of the user interface) on supplementary so that readers can get a better idea of how the platform can benefit their research. This aspect was not able to evaluate in terms of easy to use or rapid setup capability which authors claimed, because all link that authors provide was not accessible for the reviewers.

In addition to this, I have following comments and questions,

1. In the abstract and introduction, the authors mentioned that this platform ‘is suitable for a clinical study.’ Still, since this software could be classified as ‘clinical decision support’ software for deep brain stimulation surgery, authors may need to check about the regulation on this category of the software if there is no issue to describe as ‘suitable for clinical use.’

2. While the authors state that this platform provides a highly automated solution, but since there is no real demonstration of platform operation, it is not clear the level of automation that this platform is providing. For example, I am wondering interacting with other GUI software such as Paraview also fully automated and how much time could take on a regular operation task (for example, for the benchmark example) if the user has all parameters in hands.

3. The authors state that the platform offers to the user to choose an electrode geometry from the predefined collection. Does this customizable? If so, could the experimentally measured impedance spectrometer result also be used as input for estimating electrode-tissue interface?

4. While the authors mentioned that there are some limitations on performance, it is not clear what is the minimal requirement to use the platform for the demonstrated simulations (especially for the docker image). It would be helpful to the reader If the authors can provide the computing time to reproduce the same result with a sample dataset with validated hardware specs (which the hardware environment that authors operate the simulation), as supplementary.

5. Regarding the parallel computing in the frequency domain, Have the authors evaluate how much the computing time improved (by %) after parallelization compared to the serial processing? If so, do authors have any recommendations on this setting?

**Have all data underlying the figures and results presented in the manuscript been provided?**

Reviewer #1: Yes

Reviewer #2: No: All links provided by the authors were not working.

PLOS authors have the option to publish the peer review history of their article (what does this mean?). If published, this will include your full peer review and any attached files.

Reviewer #1: No

Reviewer #2: No
---

## [Decision Letter · Decision Letter 1]

6 Jun 2020

Dear Mr. Butenko,

We are pleased to inform you that your manuscript 'OSS-DBS: Open-Source Simulation Platform for Deep Brain Stimulation with a comprehensive automated modeling' has been provisionally accepted for publication in PLOS Computational Biology.

Best regards,

Daniele Marinazzo

Deputy Editor

PLOS Computational Biology

Daniele Marinazzo

Deputy Editor

PLOS Computational Biology

Reviewer's Responses to Questions

**Comments to the Authors:**

Reviewer #1: Authors have successfully addressed all concerns raised.

I would like to congratulate them for their seminal work to publish a much needed open source software for modeling complex DBS effects.

Reviewer #2: The authors have addressed all of my comments regarding the original manuscript and revised it sufficiently. Therefore, I don't have additional comments on this revised version, and I think the current manuscript has enough quality for publication in PLOS Computational Biology.

Regarding the authors' question related to the GitHub repository in the general comments, I was not able to access the linked repository with the message that the repository did not exist at the time I have been reviewing the original manuscript. This time, the issues resolved so I can access the link that the authors provided.

It seems the issue occurred because the access had tried before the authors first commit (May-11th). So, I think it would not happen for the future publication if the commit made before the review process.

Best regards

**Have all data underlying the figures and results presented in the manuscript been provided?**

Reviewer #1: Yes

Reviewer #2: Yes

PLOS authors have the option to publish the peer review history of their article (what does this mean?). If published, this will include your full peer review and any attached files.

Reviewer #1: No

Reviewer #2: No

---

## [Editor Report · Acceptance letter]

26 Jun 2020

PCOMPBIOL-D-20-00599R1 

OSS-DBS: Open-Source Simulation Platform for Deep Brain Stimulation with a comprehensive automated modeling

Dear Dr Butenko,

I am pleased to inform you that your manuscript has been formally accepted for publication in PLOS Computational Biology. Your manuscript is now with our production department and you will be notified of the publication date in due course.

With kind regards,

Matt Lyles
